# A Selective Inhibitor of Cardiac Troponin I Phosphorylation by Delta Protein Kinase C (δPKC) as a Treatment for Ischemia-Reperfusion Injury

**DOI:** 10.3390/ph15030271

**Published:** 2022-02-22

**Authors:** Nir Qvit, Amanda J. Lin, Aly Elezaby, Nicolai P. Ostberg, Juliane C. Campos, Julio C. B. Ferreira, Daria Mochly-Rosen

**Affiliations:** 1Center for Clinical Sciences Research, Department of Chemical & Systems Biology, Stanford University School of Medicine, 269 Campus Dr. Room 3145, Stanford, CA 94305, USA; nir.qvit@biu.ac.il (N.Q.); linaj@stanford.edu (A.J.L.); aelezaby@stanford.edu (A.E.); nostberg@stanford.edu (N.P.O.); julocf@gmail.com (J.C.B.F.); 2The Azrieli Faculty of Medicine in the Galilee, Bar-Ilan University, Safed 1311502, Israel; 3Department of Anatomy, Institute of Biomedical Sciences, University of Sao Paulo, Sao Paulo 05508-000, Brazil; jucruzcampos@gmail.com

**Keywords:** cardiac troponin I, cardiac ischemia-reperfusion injury, peptides

## Abstract

Myocardial infarction is the leading cause of cardiovascular mortality, with myocardial injury occurring during ischemia and subsequent reperfusion (IR). We previously showed that the inhibition of protein kinase C delta (δPKC) with a pan-inhibitor (δV1-1) mitigates myocardial injury and improves mitochondrial function in animal models of IR, and in humans with acute myocardial infarction, when treated at the time of opening of the occluded blood vessel, at reperfusion. Cardiac troponin I (cTnI), a key sarcomeric protein in cardiomyocyte contraction, is phosphorylated by δPKC during reperfusion. Here, we describe a rationally-designed, selective, high-affinity, eight amino acid peptide that inhibits cTnI’s interaction with, and phosphorylation by, δPKC (ψTnI), and prevents tissue injury in a Langendorff model of myocardial infarction, ex vivo. Unexpectedly, we also found that this treatment attenuates IR-induced mitochondrial dysfunction. These data suggest that δPKC phosphorylation of cTnI is critical in IR injury, and that a cTnI/δPKC interaction inhibitor should be considered as a therapeutic target to reduce cardiac injury after myocardial infarction.

## 1. Introduction

Myocardial infarction (MI) is the leading cause of cardiovascular death [1,2], with subsequent tissue remodeling and contractile dysfunction contributing to the development of heart failure [3,4,5,6,7]. Over the past 25 years, our lab and others have investigated the molecular response of the heart to ischemia-reperfusion (IR) injury, focusing on a specific isoform of the PKC family—δPKC. Activation of δPKC during reperfusion induces cell death through the dysregulation of mitochondrial function, apoptosis, and oncosis; a δPKC pan-inhibitor designed in our lab (δV1-1) mitigated myocardial injury by more than 60% in preclinical models of IR injury, and in patients with acute myocardial infarction [8,9]. δPKC has multiple phosphorylation targets in IR, regulating cellular metabolism and mitochondrial function, with overlapping, and often opposing, downstream effects, as exemplified by a set of substrate-specific δPKC inhibitors developed in our lab to delineate these effects [10]. Among its targets, δPKC phosphorylates cTnI at serine 23/24, serine 43/45, and threonine 144, and we have previously shown that phosphorylation of cTnI by δPKC increases with IR, and cTnI phosphorylation by δPKC is thought to cause a reduction in maximal myofilament force in cardiomyocytes [11,12]. Herein, to better understand the mechanism by which δPKC regulates IR injury, we determined the role of δPKC-mediated cTnI phosphorylation in IR using a novel selective peptide inhibitor.

## 2. Results

### 2.1. Design of a Selective Inhibitor of cTnI’s Interaction with δPKC

To determine the effects of cTnI phosphorylation by δPKC, we employed our method to rationally design a peptide inhibitor of protein–protein interactions using a homology search to identify similar regions in the otherwise non-related proteins, δPKC and cTnI (Figure 1A). We previously found that peptides corresponding to such regions of homology of otherwise unrelated, but interacting, proteins are effective inhibitors of protein–protein interactions [13]. The eight amino acid homologous sequence between δPKC and cTnI (Figure 1A) is virtually identical among multiple species (Figure 1B,C,E, and Appendix A); only conservative amino acid substitutions were observed (S to T and E to D in δPKC, and I to V in cTnI). This sequence is unique to δPKC (not present in other PKC isozymes; Figure 1D), and is exposed in both cTnI and δPKC (Figure 2A,B). An eight amino acid peptide, corresponding to the cTnI sequence (DWRKNIDA; named ψTnI) in δPKC, was synthesized as a selective inhibitor that binds δPKC to inhibit the interaction between cTnI and δPKC (Figure 2C). Using an in vitro binding assay, we found that ψTnI bound to δPKC, but not to εPKC, another PKC isozyme with high homology to δPKC, and ψTnI bound to δPKC with high affinity (K_d_ ~ 5 nM; Figure 2D,E).

### 2.2. Selective Inhibition of cTnI Phosphorylation by δPKC Prevents Ex Vivo IR-Induced Myocardial Injury

Next, we determined the effect of ψTnI on cTnI phosphorylation. Using a model of myocardial IR of isolated rat hearts (30 min ischemia followed by 60 min reperfusion; I_30min_R_60min_) in a Langendorff apparatus led to a greater than four-fold increase in the phosphorylation of cTnI, as compared to a heart subjected to perfusion under normoxic conditions (90 min perfusion). We found that this increase in cTnI phosphorylation was prevented by treatment with ψTnI at the time of reperfusion, similar to the pan-sensitive δPKC inhibitor (δV1-1) (Figure 3A). ψTnI was also specific to cTnI phosphorylation by δPKC; it had no effect on the phosphorylation of several other δPKC substrates (Figure 3B). Importantly, treatment with ψTnI at reperfusion decreased the extent of myocardial injury, as measured by creatine kinase (CK) release, an indicator of cardiac myocyte death (Figure 4A). Direct measurement of the infarct size, using a tissue stain, also showed inhibition of tissue damage (Figure 4B), to a similar extent as treatment with the pan-δPKC inhibitor δV1-1. Importantly, a peptide analog of ψTnI, in which two amino acids were substituted for an alanine (DWAANIDA; ψTnI^MU^), had no biological activity, demonstrating the specificity of the peptide (Figure 4A,B).

### 2.3. Inhibition of cTnI Phosphorylation by δPKC Attenuates Ex Vivo IR-Induced Mitochondrial Dysfunction

Mitochondrial dysfunction is a key feature of IR injury [16,17,18,19,20,21,22], and the pan-inhibition of δPKC, with δV1-1, prevents IR-induced increases in mitochondrial oxidant production, and decreases in ATP levels [23,24,25,26,27,28,29,30,31]. However, the mechanism by which δPKC causes mitochondrial dysfunction, and the protein substrate that mediates it, have not been identified. Unexpectedly, we found that ex vivo IR in rat hearts led to a decrease in the rate of maximal ADP-stimulated oxygen consumption and an increase in the rate of hydrogen peroxide production in isolated cardiac mitochondria. Inhibition of cTnI phosphorylation at the time of reperfusion improved mitochondrial function in the heart after IR; both the decreased rate of oxygen consumption and the increased rate of hydrogen peroxide production were prevented with ψTnI treatment (Figure 4C,D). 

## 3. Discussion 

Cardiac troponin I regulates cardiomyocyte contraction in conjunction with troponin C and troponin T via the calcium-mediated interaction between actin and myosin. In an ex vivo model of cardiac IR injury, we observed a decrease in myocardial damage, as evidenced by a reduction in creatine kinase release and infarct size when the δPKC-mediated phosphorylation of cTnI was selectively inhibited by a novel protein–protein interaction inhibitory peptide, ψTnI; the effect of ψTnI was similar to the benefit observed when treating with the pan-inhibitor of all δPKC phosphorylation, δV1-1. In addition to attenuating IR-induced myocardial injury, blocking the phosphorylation of cTnI by δPKC ameliorated the mitochondrial dysfunction associated with IR, as evidenced by the increased mitochondrial oxygen consumption and decreased mitochondrial oxidant production. 

Myocardial infarction is the leading cause of cardiovascular mortality, with injury occurring during ischemia and subsequent reperfusion. Most therapeutics are targeted towards re-establishing and maintaining perfusion, and there are currently no established therapies to mitigate reperfusion injury. We previously found that δPKC is a key mediator of IR injury, and the inhibition of δPKC activity, with the pan-inhibitor peptide δV1-1, at the time of reperfusion attenuates injury in animal models of IR, and in patients with acute myocardial infarction [8,9]. δPKC has multiple substrates with various, and often opposing, downstream effects [10,32]. cTnI is known to be phosphorylated by δPKC in IR [10,11,12]. In vitro studies suggested that cTnI phosphorylation by δPKC impairs cardiomyocyte shortening [12,33,34,35], and transgenic expression of mutant cTnI, lacking δPKC phosphorylation sites (Ser-43, -45->Alanine), in mice improves cardiac performance [36]. Here, we demonstrate that the targeted inhibition of cTnI phosphorylation by δPKC, specifically in IR injury, at the time of reperfusion, ameliorates IR injury and IR-induced mitochondrial dysfunction, suggestive of a deleterious role of this phosphorylation event in IR. 

An IR event leads to critical changes in mitochondrial function, bioenergetics, and redox balance in the cardiomyocyte [18,37]. These mitochondrial changes result in cell death, tissue remodeling, and contractile dysfunction [21]. Here, we measured the rates of oxygen consumption and production of the oxidant species H_2_O_2_ from isolated cardiac mitochondria after IR. As expected, cardiac IR decreased mitochondrial function (as measured by oxygen consumption), and increased H_2_O_2_ production. Interestingly, treatment with ψTnI was sufficient to prevent the IR-induced increase in mitochondrial oxidant production and the decrease in oxygen consumption. This improvement in mitochondrial function was paralleled by an improvement in tissue viability in IR hearts treated with ψTnI. 

We rationally designed and utilized a selective peptide inhibitor of this phosphorylation event as a tool to assess the role of cTnI phosphorylation in IR injury. The advantage of our approach is that peptides have good biodistribution, with cell entry facilitated by the conjugated TAT sequence; peptides are water-soluble and consist of naturally occurring amino acids [38]. Furthermore, the cellular interactome has been predicted to contain about 650,000 protein–protein interactions (PPIs) [39,40,41,42]. Peptides and peptidomimetics (modified peptide analogs that mimic the structural and functional characteristics of peptides in 3D space) can be harnessed to target PPIs, as about 15–40% of all PPIs are mediated by a short linear peptide [43,44,45,46]. Since PPIs regulate critical molecular communications, peptides that target PPIs have significant therapeutic potential and an excellent in vivo safety profile [47]. For IR experiments, isolated rat hearts were treated with 1µM of peptide because of poor peptide pharmacokinetics; with multiple arginine and lysine residues, these peptides are quickly degraded by proteases. Dosing was guided by prior dose–response studies, using the δV1-1 peptide [48]. In summary, we describe the development and design of a highly potent selective peptide inhibitor of cTnI phosphorylation by δPKC. Inhibition of this phosphorylation event has clear beneficial implications in IR injury and sheds light on the role of cTnI phosphorylation in IR injury. 

## 4. Materials and Methods

### 4.1. Animals

Between 10- and 12-week-old wild-type male Sprague Dawley rats (250–300 g) were purchased from Charles River; only male rats were used to reduce the potential confounding factor of cardioprotective estrogen effects. Rats were anesthetized by intraperitoneal injection of sodium pentobarbital (30–90 mg/kg) and sodium heparin (1000 units/kg). After 4–5 min, pain sensitivity was performed via a toe pinch, and the hearts were removed by a terminal surgical procedure. The Administrative Panel on Laboratory Animal Care at Stanford University (protocol #10363) and the Ethic Committee on Animal Use of the Institute of Biomedical Sciences at University of Sao Paulo (protocol #60/2017) approved all animal protocols.

### 4.2. Peptide Synthesis

Peptides were synthesized as previously described [10]. In brief, ψTnI peptide and its analog ψTnI^MU^ peptide were synthesized on solid support using a fully automated microwave peptide synthesizer (Liberty, CEM Corporation), using a homology sequence, conjugated to TAT_47-57_ carrier peptide, a short positively charged peptide that is used as a carrier for the delivery of the peptide into the cell, using a Gly-Gly spacer. The peptides were synthesized by SPPS (solid phase peptide synthesis) methodology with a fluorenylmethoxycarbonyl (Fmoc)/tert-butyl (tBu) protocol. The final cleavage and side chain deprotection was performed manually without microwave energy. Peptides were analyzed by analytical reverse-phase high-pressure liquid chromatography (RP-HPLC) (Shimadzu, MD, USA) and matrix-assisted laser desorption/ionization (MALDI) mass spectrometry (MS), and purified by preparative RP-HPLC (Shimadzu, MD, USA).

### 4.3. Peptide Binding to δPKC and εPKC in Vitro

In vitro binding assay was performed as previously described [10]. Peptides (6 μM) were covalently attached by the amine of the peptide to the carboxyl on the graphene sensor chip. After rinsing in PBS, baseline current levels for the chip were recorded on AGILE Device label-free binding assay (Cardea Inc., San Diego, CA, USA). Next, PBS was aspirated, and a 50 μL droplet of the tested protein, at 75 µg/mL, was applied to the chip and the change in the sensor chip readout was recorded. After data were gathered, the responses of 25 sensors were averaged, and any background drift recorded in PBS was subtracted. A Hill equation fit was used to determine K_d_. K_d_ values were also calculated by measurement of the K_on_ and K_off_ values at a single concentration. This was performed by fitting the binding curve to a double exponential function, and the first rinse to a single exponential, using a single concentration. Results are from three independent experiments.

### 4.4. Cardiac Ischemia Reperfusion—Ex Vivo Langendorff Model

Isolated rat hearts were hung on a Langendorff apparatus in the Mochly-Rosen laboratory at Stanford University, as we previously described [10]. Only rat hearts hung in less than 1 min were used for experiments; this was to minimize the possibility of pre-conditioning effects. In addition, the order of control and experimental rat heart conditions was randomized each day the Langendorff was conducted. Krebs–Henseleit solution was perfused through the heart on the apparatus for 10 min to allow the heart to equilibrate to the system. For IR experimental conditions, all perfusion was stopped (ischemia) for 30 min and restarted (reperfusion) for 60 min. Used as a control, normoxia conditions were flowed for 100 min to match the total time the ischemia-reperfusion hearts were hung. For experimental conditions including ψTnI treatment, the peptide was added to the buffer, perfusing the heart for the first 20 min of reperfusion. Perfusate was collected for a creatine kinase (CK) assay up to 30 min into reperfusion, as well as the matching time for normoxia, and ran as previously described [10]. CK assay was used to confirm that increased cell death had occurred as expected with IR compared to normoxic conditions. Furthermore, after the hearts were taken down, TTC (2,3,5-TTC-triphenyltetrazolium chloride) staining was performed with a slice of the heart; the same slice was used for each heart and incubated in 1% fresh TTC at 37 °C for 15–20 min, protected from light. After incubation, the heart slices were washed with fresh 10% formalin solution. A scanner was used to image each slice and ImageJ software was used to quantify viable tissue relative to the total area. The remainder of the heart was used for mitochondrial isolations, as previously described [49]. Treatment with TAT (vehicle), δV1-1 (TAT-δV1-1) [8], ψTnI (TAT-DWRKNIDA) or ψTnI^MU^ (TAT-DWAANIDA) was administered at 1μM at the time indicated in Figure 4A. δV1-1 was synthesized by KAI Pharmaceuticals (South San Francisco, CA) and stored at −80 °C as dry powder until use.

### 4.5. Molecular Docking of cTnI in δPKC

PDB ID: 4Y99 was loaded into the Molecular Operating Environment (MOE) 2019.0102 software; 4Y99.C chain (cTnI) was kept and other chains were deleted. 4Y99.C was prepared using the QuickPrep functionality in MOE with default settings. QuickPrep optimizes the H-bond network and performs energy minimization on the system.

### 4.6. Computational Analyses

In order to better visualize the degree of conservation of cTnI across evolution, a heat map was constructed. A BLAST query was performed on the ψTnI sequence and the top 50 sequences, scored by similarity, were collected. Then, orthologs for each species were collected. For each gene, the number of identities compared to the human ψTnI sequence was enumerated. These data were then expressed as a heatmap. To further examine the degree of conservation of the ψTnI sequence, a PSI-BLAST (Position-Specific Iterative BLAST) was performed using human cTnI (UniProt ID: P19429) and human δPKC (UniProt ID: Q05655) as the query sequences. PSI-BLAST was utilized as it utilized more general protein “features” to identify homologies; therefore, it is better suited to examine distant evolutionary relationships [50]. The top 500 hits from each query were collected and analyzed. At each position, the number of similar and identical residues to the human gene was summed, meaning that a score of 500 at a position represents perfect conservation. 

### 4.7. Western Blot Analysis

A Bradford assay (Bio-Rad Protein Assay Dye Reagent Concentrate (#5000006)) was used for protein quantification, after samples were lysed in 1x lysis buffer, and lysate was separated from cell debris (21,000× *g* 15 min at 4℃). Samples were boiled with Laemmli buffer containing 2-mercaptoethanol, loaded on SDS–PAGE, and transferred to PVDF membrane, 0.45 μm (Bio-Rad, Hercules, CA). Membranes were probed with the antibody of interest, and then visualized by ECL (0.225 mM p-coumaric acid; Sigma, St. Louis, MO), 1.25 mM 3-aminophthalhydrazide (Luminol; Fluka) in 1 M Tris pH 8.5. Antibody specificity was validated based on the target protein molecular weight on Western blot. The following antibodies were used in this study: cardiac troponin I (Abcam, Cambridge, UK; ab47003; 1:500; lot #: GR3248433-1); troponin I-C antibody (D-12) (Santa Cruz, Dallas, TX; sc-31655; 1:100; lot #: F3015); phospho-troponin I (Cardiac) (Ser23/24) antibody (Cell Signaling, Danvers, MA; #4004; 1:500; lot #: 3); phospho-IRS1 (Cell Signaling 2381); phospho-Drp1 (Cell Signaling 3455); phospho-MARCKS (Cell Signaling 2741); phospho-STAT (Cell Signaling 8826S); mouse IgG HRP linked whole Ab (Sigma; #NA931-1ML, 1:5000; lot #: 17041904); rabbit IgG HRP linked whole Ab (Sigma; #NA934-1ML, 1:5000; lot #: 17065614).

### 4.8. Mitochondrial Function

Heart mitochondrial function was assessed as previously described [49]. Briefly, cardiac samples were minced in isolation buffer (300 mM sucrose, 10 mM HEPES, 2 mM EGTA, pH 7.2, 4 ℃ ), containing type I protease (bovine pancreas; Sigma, P4630), centrifuged at 500× *g* to pellet cell debris, followed by centrifugation of supernatant at 9000× *g*, resulting in a mitochondrial-enriched pellet, which was resuspended in a minimal volume of isolation buffer. Mitochondrial O_2_ consumption and H_2_O_2_ release were measured in 0.125 mg/mL of isolated mitochondria in experimental buffer (125 mM sucrose, 65 mM KCl, 10 mM HEPES, 2 mM inorganic phosphate, 2 mM MgCl_2_, 100 μM EGTA, 0.01% BSA, pH 7.2), containing succinate 2 mM (Sigma, S3674), malate 2 mM (Sigma, M1000) and glutamate 2 mM (Sigma, G1251) substrates, with continuous stirring at 37 °C. ADP 1 mM (Amresco, 0160) was added to induce state 3 respiratory rate. Mitochondrial O_2_ consumption was monitored using a computer-interfaced Clark-type electrode (OROBOROS Oxygraph−2k). Mitochondrial H_2_O_2_ release was measured using Amplex Red 25 μM (Molecular Probes A12222) horseradish peroxidase 0.5 U/mL (Sigma P8125) system, and detected using a fluorescence spectrophotometer (ʎex = 563/ʎem = 587 nm) (F-2500 Hitachi—Hitachi).

### 4.9. Statistical Analysis

Values are presented as mean ± standard deviation relative to the average value for the control group, unless otherwise stated. Group differences were assessed by one-way ANOVA with Tukey’s correction for multiple comparisons with statistical significance established at *p* < 0.05 (GraphPad Prism).

## 5. Conclusions

We rationally designed and utilized a selective peptide inhibitor of δPKC phosphorylation of cTnI as a tool to assess the role of cTnI phosphorylation during IR injury. The inhibition of cTnI phosphorylation by δPKC with ψTnI attenuates myocardial IR injury and prevents IR-induced mitochondrial dysfunction.

## Figures and Tables

**Figure 1 pharmaceuticals-15-00271-f001:**
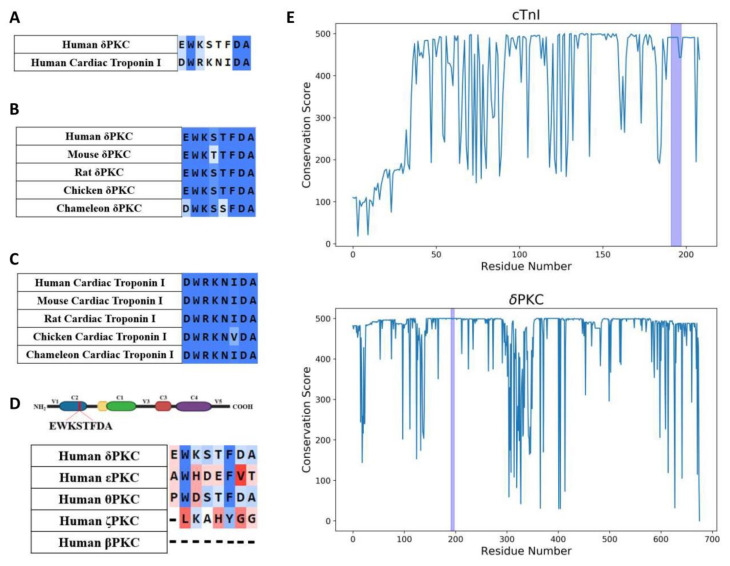
ψTnI sequence is highly conserved. Sequence alignment identifies a short sequence of homology between the C terminus of human cTnI and δPKC C2 domain (**A**) that is conserved in evolution (**B**,**C**), but is absent in other PKC isozymes (**D**). In (**E**), conservation is measured by the number of identical or similar residues compared to the human cTnI (**top**) and δPKC (**bottom**) gene. Blue regions represent the sequences corresponding to ψTnI in cTnI and δPKC, respectively.

**Figure 2 pharmaceuticals-15-00271-f002:**
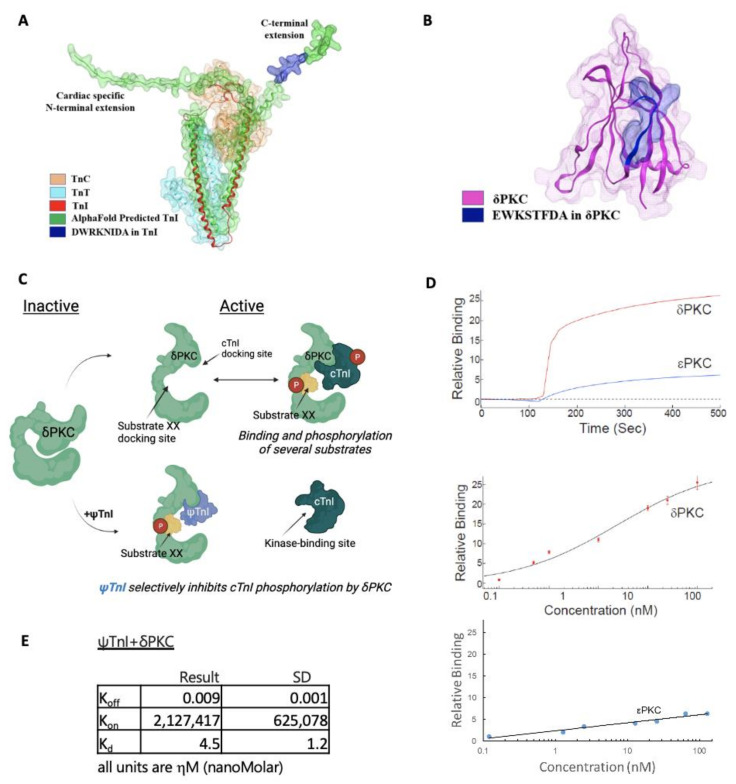
Binding activity and selectivity of ψTnI in vitro. The ψTnI sequences (blue surface) are exposed in the C terminus unstructured region of cTnI (**A**), as observed in the AlphaFold predicted structure of cTnI (AF-P19429-F1) [14] shown in the green surface superimposed with troponin I in the crystal structure of the complex of troponin T (cyan surface), C (orange surface), and I (red ribbon) (PDB ID: 4Y99) [15]; they are also exposed in the C2 domain (highlighted in pink; (**B**)) in δPKC. (**C**) A scheme describing the mechanism of an inhibitor that selectively inhibits the docking and phosphorylation of cTnI by the multi-substrate kinase δPKC. ((**C**); top) Intramolecular interactions in δPKC are disrupted by PKC activation, exposing the catalytic site and selective substrate docking sites on δPKC. Docking of these substrates to the kinase, concomitantly or one substrate at a time, increases the access of the catalytic site for the substrates, leading to their phosphorylation (red P circles). In the inactive δPKC (left), the docking site interacts with a PKC sequence, e.g., ψTnI site, which mimics the kinase docking site on cTnI. ((**C**); bottom). ψTnI is a competitive inhibitor of docking and phosphorylation of cTnI by δPKC; it does not affect docking and phosphorylation of other δPKC substrates (e.g., substrate XX, yellow). (**D**) Binding curves of δPKC and εPKC, at 75 µg/mL (about 1 μM), to ψTnI (6 µM). The peptide selectivity binds to δPKC, and not to another novel PKC, εPKC. Binding assay with increasing amounts of δPKC to ψTnI and binding assay with increasing amounts of εPKC. (**E**) Calculated K_off_, K_on_, and K_d_ between ψTnI and δPKC, shown in ηM. Results are from three independent experiments.

**Figure 3 pharmaceuticals-15-00271-f003:**
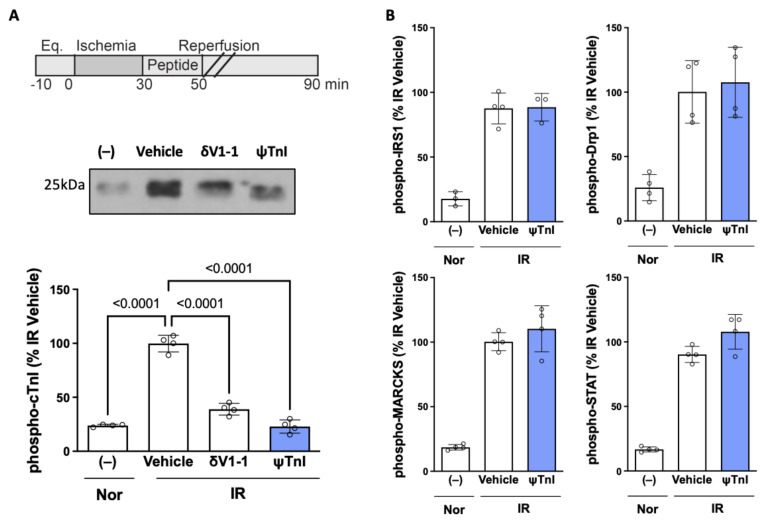
ψTnI inhibits cTnI phosphorylation by δPKC. Male rat hearts were subjected to 90 min perfusion (Nor; normoxia) or I_30min_R_60min_ in the presence of a selective pan-inhibitor of δPKC (δV1-1), control (vehicle; TAT), or in the presence of the selective inhibitor of cTnI, ψTnI. Phosphorylation was determined by Western blots. ψTnI selectively inhibits δPKC-mediated cTnI phosphorylation (**A**). ψTnI does not affect phosphorylation of other δPKC targets, such as dynamin-related protein 1 (Drp1), signal transducer and activator of transcription (STAT), insulin receptor substrate 1 (IRS1), and myristoylated alanine-rich c-kinase substrate (MARCKS) (**B**); n = 4 biological replicates (open circles) unless otherwise stated; 1µM δV1-1, TAT, or ψTnI used. Data were evaluated by one-way ANOVA with Tukey’s multiple comparisons between each treatment group and *p* < 0.05.

**Figure 4 pharmaceuticals-15-00271-f004:**
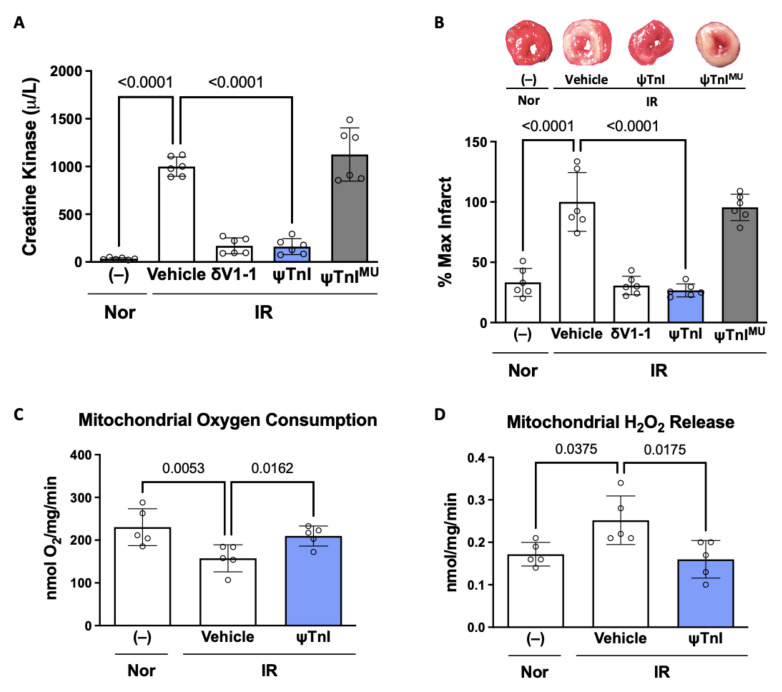
ψTnI prevents IR-induced cardiac damage and mitochondrial dysfunction. Treatment with ψTnI attenuates cardiac injury following IR in male rat hearts, similar to the pan-δPKC inhibitor δV1-1, while ψTnI^MU^, an inactive analog of ψTnI, does not, as demonstrated by creatine kinase release (**A**) and infarct size (**B**); n = 6/condition. IR-induced mitochondrial dysfunction in male rat hearts was attenuated with ψTnI treatment at reperfusion, as evidenced by changes in rates of maximal ADP-stimulated (state III) oxygen consumption (**C**) and H_2_O_2_ production (**D**); n = 5 biological replicates unless otherwise stated; 1µM δV1-1, TAT, ψTnI, or ψTnI^MU^ used. Data were evaluated by one-way ANOVA with Tukey’s multiple comparisons between each treatment group and *p* < 0.05.

## Data Availability

Data is contained within the article or Appendix A.

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
