# Peer review of "A Selective Inhibitor of Cardiac Troponin I Phosphorylation by Delta Protein Kinase C (δPKC) as a Treatment for Ischemia-Reperfusion Injury"

_pharmaceuticals, 2022, doi:10.3390/ph15030271_

Round 1

Reviewer 1 Report

The authors rationally designed a novel peptide inhibitor that selectively interferes with the interaction of cardiac troponin I with protein kinase C-delta. The authors used this peptide to investigate the role of troponin I phosphorylation in the prevention of cardiac injury in the Langendorf model of heart attack, ex vivo. Moreover, this peptide protects mitochondria upon cardiac injury. This exciting work shows the promising effects of a novel peptide inhibitor in reducing cardiac injury.

Author Response

Thank you for the opportunity to submit our revised manuscript entitled “A Selective Inhibitor of Cardiac Troponin I Phosphorylation by Delta Protein Kinase C (δPKC) as a Treatment for Ischemia-Reperfusion Injury” to Pharmaceuticals. We appreciate the efforts of the editorial team and the insightful comments of the reviewers.

Reviewer 2 Report

  1. Add a table for Kon and Koff values of ψTnI for δPKC and εPKC
  2. In the upper panel of Figure 2D, what concentration was used in the binding assay of ψTnI for δPKC and εPKC?
  3. If was done binding of ψTnI for εPKC, the authors should add the curve
  4. The FEB technology is superior to other techniques such as SPR for binding assays? This should be discussed.
  5. Which is the Kd of δV1-1 for δPKC and how the concentration (1 mM) used in IR experiments were determined, is there a correlation of the Kd value (5 nM) for ψTnI inhibitor with the concentration used in these assays? This should be discussed at least.
  6. Which is the supplier of the δV1-1 inhibitor?
  7. If the IR experiments were performed at Sao Paulo University should be stated. It is unclear where these studies were conducted.

Author Response

(The authors gave the same response as above.)

Reviewer 3 Report

To:

Editorial Board

Pharmaceuticals

Title: “A Selective Inhibitor of Cardiac Troponin I Phosphorylation by Delta Protein Kinase C (δPKC) as a treatment for Ischemia-Reperfusion Injury”

Dear Editor,

I read this paper and I think that:

  • I think that the authors should provide a graphical abstract which should represent the mechanism of action of their selective inhibitor. This would improve the readability of the text.
  • the authors should provide data about the amount in ROS induced by ischemia/reperfusion and the differences in concentrations among groups.
  • The most intriguing effect of ischemia-reperfusion injury is the impact on mitochondria and their function. Alterations in mitochondria might impact on the survival of cardiac cells. Can the authors provide data on the impact of their selective inhibitor on the function of mitochondria on injured cells?

Author Response

(The authors gave the same response as above.)

Round 2

Reviewer 3 Report

the authors well addressed my previous comments. The paper improved very much